# Triggering Influence of Seasonal Agricultural Irrigation on Shallow Loess Landslides on the South Jingyang Plateau, China

**Rui-Xin Yan [1,2,3], Jian-Bing Peng [1,2,\*], Qiang-Bing Huang [1,2], Li-Jie Chen [2], Chen-Yun Kang [2] and Yan-Jun Shen [4]** 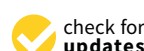

[1] Key Laboratory of Western Mineral Resources and Geological Engineering of Ministry of Education, Chang'an University, Xi'an 710054, China

[2] College of Geology Engineering & Geomatics, Chang'an University, Xi'an 710054, China

[3] College of Architecture and Civil Engineering, Xi'an University of Science and Technology, Xi'an 710054, China

[4] College of Geology and Environment, Xi'an University of Science and Technology, Xi'an 710054, China

\* Correspondence: dicexy_1@126.com; Tel.: +86-139-9288-1198

**Abstract:** Since large-scale agricultural irrigation began in the 1980s, 92 landslides have occurred around the South Jingyang Plateau during the past 40 years. The geological disaster and soil erosion have caused numerous casualties and substantial property loss. In this work, several field investigations are carried out to explore the soil erosion and mechanical mechanism of these irrigated shallow loess landslides on the South Jingyang Plateau. (1) We investigated the spatial distributions, types and developmental characteristics of loess landslides. (2) We surveyed and monitored seasonal agricultural irrigation features and groundwater changes in the area since the 1980s and found that irrigation is a significant factor influencing groundwater changes, soil erosion and even causing landslides to occur. (3) Based on the field investigation, the occurrence of these irrigated shallow loess landslides was generalized, and it was found that the core process was due to the liquefaction of softening zone. We carried out a static liquefaction test and verified that the natural loess was prone to liquefaction. (4) The three main reasons for shallow loess landslides in the South Jingyang Plateau were discussed. This study provides a valuable reference for achieving an understanding of the relationship between seasonal agricultural irrigation and the occurrence of loess landslides in the area as well as similar irrigated agricultural areas.

**Keywords:** loess landslide; agricultural irrigation; field investigation; static liquefaction; soil erosion

## 1. Introduction

The loess plateau, a special geomorphologic landscape located in the northwestern area of China, covers approximately 4.4% (620,000 km$^2$) of China's total land area [1,2]. Due to the continental monsoon climate limit, seasonal irrigation has become a necessary approach for maintaining agricultural production in this area [3]. Loess has typical features including macro-pores, vertical joints, weak cementation and high sensitivity to water [4–6]. These characteristics are prone to lead to soil erosion, structural collapse under long-term immersion water and may even lead to instability [7–9]. Thus, seasonal agricultural irrigation has become an important triggering factor inducing serious landslides on the loess plateau of northwest China [10–12]. As a matter of fact, agricultural activity has become an important threat to the natural process around the world. For example, man-made terraces in many areas of the world lead to an increase in instability hazards according to the previous reports [13–15]. Irrigation concrete-ditch prevented the evaporation of groundwater, resulting in the

"canopy-cover" effect [16,17]. Focusing on the loess plateau, many loess landslides caused by seasonal irrigation have been also occurred [2,18–20]. For instance, there have been 120 loess landslide events in the Heifangtai area, which have killed approximately 70 people, due to irrigation, since 1984 [21]. In addition, the phenomenon of agricultural irrigation influencing the local climate change should be also investigated [22–24]. Therefore, it is necessary to carry out the systematic study on the influence of seasonal agricultural irrigation on natural changes in the loess plateau.

In the study area of the south Jingyang platform of Shaanxi Province, China, the large-scale water irrigation performed in farmland since the 1980s has caused the loess to suffer the soil erosion under its own weight, which has damaged buildings, and slope instability has occurred due to infiltration of irrigation water in some areas. According to statistics, there have been as many as 92 multi-cyclical landslides in this region, resulting in the fact that at least 17 places have experienced at least two slides, and more than 30 people have been reported dead. One of these landslides occurred on 2 December 1984, killed 20 people, and led to 20 people being seriously injured. In addition, a high, steep slope has formed at the edge of the south Jingyang platform in Shaanxi Province, due to the considerable lateral erosion near the Jinghe River. A pumping station built along the river increased the irrigation volume and seriously affected the stability of the slope, resulting in frequent landslides along the edges of the plateau [25]. With tableland farmland collapsing and declining, the cultivated farmland area has decreased continuously, in turn, the development of local agriculture has been seriously affected and great property losses have been caused to farmers on the plateau. In addition, Xianyang International Airport in the largest city of Xi'an in the Northwest in China is located on the tableland, and the change in the drainage system on the raft due to soil erosion will also affect normal operations of the airport [26].

The mechanism of the loess landslides on the south Jingyang platform has been studied by many authors due to their disastrous effects [25–28]. Some authors hypothesize that irrigation water infiltrates into the slope along vertical joints, which leads to an increase in the degree of saturation of the soil. The local loess then becomes saturated, and instability occurs [11,20]. At present, studies on the mechanism of irrigation-induced loess landslides are mostly carried out through approaches such as laboratory tests, field tests, and numerical simulations. For example, Puri [29], Zhang et al. [10], Li et al. [27], Xu et al. [19] and Li et al. [18] discussed how irrigation induces loess landslides step by step through a series of indoor triaxial tests and stated the generation of pore water pressure and shear deformation during loess saturation. These authors suggested that with the infiltration of irrigation water into the slope, the perched water level rises, leading to failure to the upper slope and then fluidization, which induces high-speed and long-distance landslides. Here, the key point is that the liquefaction of loess is an important cause of rapid, long-distance loess flow slides induced by irrigation. Cui et al. [30] adopted a centrifugal model test to simulate the mechanical behavior of the loess slope on the Heifangtai platform under irrigation conditions. The characteristics of loess slope deformation, pore water pressure and soil pressure were systematically tested, and the characteristics of the evolution of the instability of the loess slope caused by irrigation were discussed. It has been inferred from laboratory shear tests that static liquefaction of the loess slope foundation is one of the main reasons for the instability of the loess slope [29–31].

In addition, the ring shear test has been applied to test the undrained shear characteristics of loess to some extent. Peng et al. [26] carried out a consolidated undrained shear test and ring shear test, revealing the characteristics and mechanism of loess sliding-flow landslides on the south Jingyang platform. It was concluded that the surface friction of landslides caused by irrigation was deeper than that of landslides caused by the Wenchuan earthquake and that saturated loess samples exhibited a high liquefaction ability. Zhang et al. [32] carried out undrained ring shear tests on loess samples with different initial porosities, hypothesizing that soil densification caused by irrigation may lead to the occurrence and mobility of landslides along the plateau. Some authors have also carried out irrigation simulation tests using the farmland field conditions or similar indoor models to analyze the effect of irrigation on the stability of tableland slopes. Xu et al. [11] conducted laboratory tests and large-scale

field tests on a typical cracked plateau in Heifangtai to simulate the impact of irrigation on slope stability. The test showed that cracks had a significant impact on the irrigation water flowing into the ground. With the infiltration of irrigation water into the main cracks, the pore water pressure on the loess slope increases rapidly, causing local instability of the slope. Numerical simulations of irrigation-induced landslides in loess areas have also been studied to some extent. For example, Lian et al. [33] applied Phase2 software to numerically simulate the typical profile of Huangci landslides, considering the stability evolution law of landslides under conditions of a long-term rise and fall of irrigation water levels, and these authors revealed the general law of irrigation-induced landslides in this area through numerical simulation. Similarly, Pan et al. [34] used a numerical simulation method to analyze the failure mechanism of loess landslides on the Heifangtai platform caused by irrigation. Li et al. [27] applied the improved Sassa *K* model to simulate the sliding of landslides.

In general, previous studies on agricultural irrigation-type landslides in this region have mostly adopted approaches such as indoor tests, field tests, and numerical simulations. By carrying out tests and simulations on the relationship between the mechanical properties of loess and water, the induced effects of irrigation on loess landslides are indirectly reflected. This study focuses on a systematic investigation of the hydrogeological conditions of loess landslides in this area, through statistical analysis of the relationship between irrigation, groundwater and intuitively establishes the internal relationship between agricultural irrigation and landslides. Then, a generalized model of landslide occurrence is proposed to reveal the internal mechanism of shallow surface landslides induced by seasonal agricultural irrigation in this region.

## 2. Materials and Methods

### 2.1. Geographic, Geomorphologic and Climatic Features on the South Jingyang Plateau

The South Jingyang Plateau, located on the south bank of Jinghe River, in Jingyang county, Shaanxi Province, extends from east to west in terms of overall direction, with a total extension of 28 km, mainly passing through the three administrative townships of Taiping, Jiangliu and Gaozhuang (shown in Figure 1). The Quaternary loess was deposited on the southern uplift of the Jinghe River, influenced by the hidden faults of Jinghe River, and eventually formed the loess tableland whose area is approximately 70 km$^2$. The difference in elevation between its top and bottom is approximately 30–90 m, and the elevation is approximately 420–490 m above sea level. The tableland surface is open and flat, suitable for cultivation. In addition, due to the effect of long-term lateral erosion of the Jinghe River, the edge of the tableland is 30–90 m high, and the slope is 45°–80° [25]. The south Jingyang platform has been affected by large-scale agricultural irrigation since 1980 and had experienced 92 loess landslide accidents as of April 2016. Among these events, many slips have occurred in 17 places, causing numerous casualties and substantial property losses [35].

Furthermore, Jingyang belongs to a temperate continental monsoon climate zone with four distinct seasons and average annual precipitation of 548.7 mm, but rainfall is extremely uneven. The annual precipitation is mainly concentrated in July–September, accounting for 71% of the total annual precipitation. Figure 2 shows the monthly average precipitation distribution in the region from 2000 to 2017. Rainfall in July–September is relatively heavy, exceeding 80 mm, and especially that in July and September is over 90 mm. In other months, rainfall is significantly lower. From December to February of the second year, the monthly average precipitation did not exceed 15 mm. To ensure the normal operation of agricultural production, irrigation has become an inevitable choice.

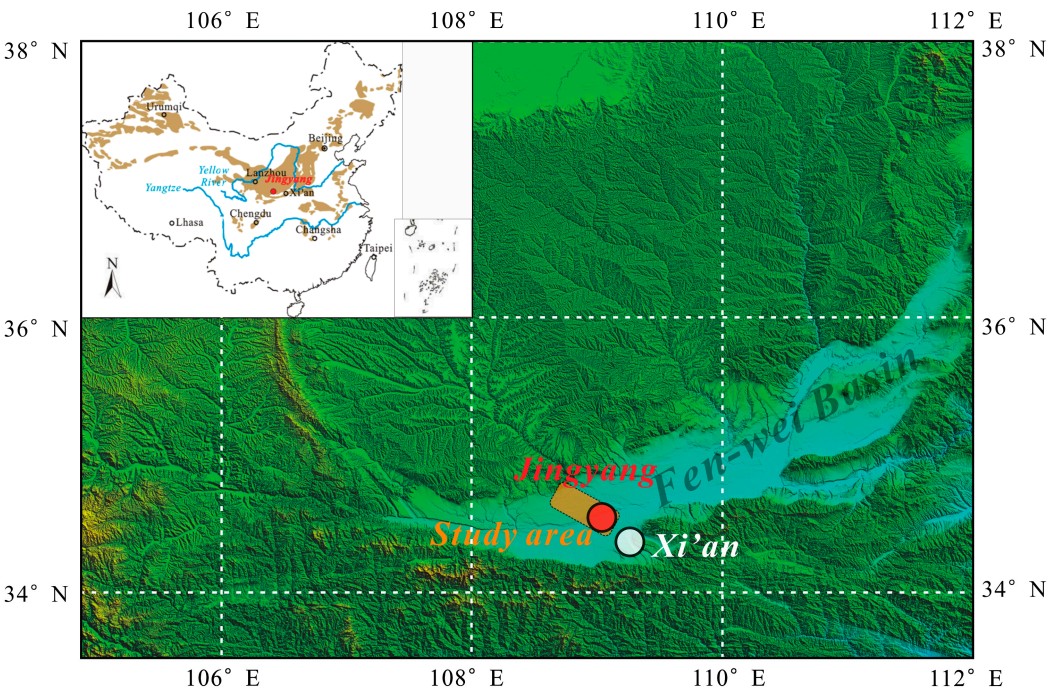

**Figure 1.** Study location of the South Jingyang Plateau in the northwestern of China.

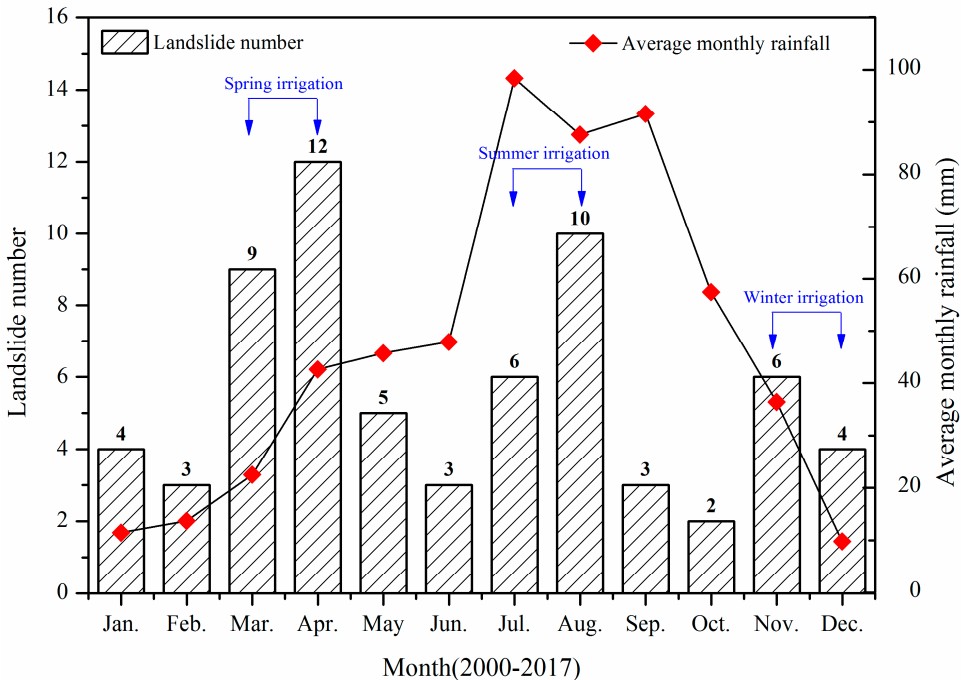

**Figure 2.** Monthly average precipitation distribution on the South Jingyang Plateau from 2000 to 2017.

## 2.2. Landslide Development Survey on the South Jingyang Plateau

To further understand the developmental characteristics of landslides in the region, a detailed survey of the development characteristics of the different landslide groups in the region was carried out, and the following main features were found. Due to the strong lateral erosion of the Jinghe River, the slope of the trailing edge of the landslide group is generally steep. The average slope of the trailing edge of the landslide group can reach 50°, and the earlier it slides, the steeper the slope of the corresponding trailing edge. The steep slope along the edge of the platform creates appropriate gravity conditions for the formation of a landslide. There are many cracks after a landslide occurs

on tableland. The cracks are mostly arc-shaped, and the overall trend is parallel to the side of the plateau. Such cracks are concentrated in the areas where landslides occur frequently (such as Miaodian or Jiangliu), and the penetration is good. According to statistics, there are 36 cracks in the study area with a length of more than 5 m. The distance between a crack and the edge is generally less than 30 m, accounting for more than 90% of the total number of cracks. The farther away from the edge, the lower the number of cracks. Some cracks show vertical dislocation. Parts of cracks are accompanied by sinkholes. The cracks are formed by the enrichment of irrigation water to form water holes of different scales. A landslide is mostly cut out from the slope foot, and it then slides along the terrace over a long distance, potentially extending for 200–300 m. The resulting landslide body is mostly tongue-shaped, within which depression of the trailing edge is "crescent-shaped". That is, the ends are narrow and shallow, and the center is wide and deep. In the center, the drums and the depressions are arranged to form a wavy terrain, and the drums and depressions extend in a direction perpendicular to the sliding direction. Most of the landslide front is accompanied by mud extrusion, which is like the mudflow accumulation, and the leading edge tends to extend over a long distance.

### 2.3. Agricultural Irrigation Survey on the South Jingyang Plateau

According to the agricultural production rules of this region, it can generally be divided into three stages of irrigation. Spring irrigation: It is concentrated from early March to mid-April every year. The spring crops recover and grow, and rainfall is obviously insufficient. To ensure the growth of spring seedlings, spring irrigation is mostly provided by diverting water from the Jinghe River. Summer irrigation: It is concentrated from early July to late August each year. Although rainfall is high at this stage, it is not uniform, and most rainfall events are concentrated and drastically reduced. At this stage, the growth of crops is nearing the harvest stage, and the water requirement is high. In addition, evaporation in summer is high, and summer irrigation is needed in Jingyang. According to historical data, summer evaporation on the South Jingyang Plateau is greater than 150 mm. The amount of evaporation in summer accounts for more than half of the total annual evaporation. The maximum evaporation per month in summer can reach 275 mm [36]. The amount of rainfall in summer is much less than the amount of evaporation. Due to a large amount of evaporation in summer, irrigation contributes less to the rise of the groundwater level. Winter irrigation: Due to the freezing away effect after winter irrigation, the surface soil forms 1–2 cm-thick agglomerates, which can reduce evaporation from the ground and loosen the soil. This situation is conducive to the storage of water. Therefore, winter irrigation is often carried out on the South Jingyang Plateau. Winter irrigation is mainly concentrated in early November to late December. However, considering that the region is cold in winter, irrigation water tends to freeze in the ground, and the maximum frozen soil depth can reach 44 cm [37]. Therefore, the actual infiltration water volume and depth will be affected by freezing.

At present, current agricultural irrigation in the region is mostly based on traditional flood irrigation. Because the topography of the tableland side is slightly tilted toward the inner area of the tableland, to facilitate the flow of water in a ditch, the main channel on the South Jingyang Plateau is arranged along the rim. Due to leakage of the main canal and broad irrigation, groundwater recharging on the plateau is artificially increased, which provides important conditions for the occurrence of landslides.

### 2.4. Experimental Method of Loess Liquefaction Tendency Due to Irrigation

To explore the effect of irrigation on the slope, we also systematically carried out static liquefaction tests on typical soil samples (from middle Pleistocene Q2 undisturbed loess) from the South Jingyang Plateau. In this work, all of loess samples were collected from the investigation adit located in the rear of Zhaitou landslide. The buried depth of the adit is about 22 m from the tableland roof. In order to avoid the disturbance of natural loess, each sampled natural loess was trimmed to a cuboid-shaped block with 30 cm × 20 cm at the sample site. The block was cut into a cylindrical sample with a diameter of 12 cm and a height of 15 cm and sealed in a steel cylindrical bucket by wrapping tape and liquid paraffin to avoid water desorption. Each one was cut a standard cylinder as natural undisturbed loess sample with

a diameter of 5 cm and a height of 10 cm in the laboratory. The basic physical parameter tests including moisture content, dry density, porosity ratio, plastic and liquid limits of the prepared loess samples were performed firstly respectively. The measured basic physical parameters are listed in Table 1.

**Table 1.** Basic physical parameters of Q2 loess samples from a landslide located in the South Jingyang Plateau.

| Sample Site | Natural Moisture Content (%) | Dry Density (g/cm³) | Porosity Ratio | Specific Gravity | Liquid Limit (%) | Plastic Limit (%) |
|---|---|---|---|---|---|---|
| Q2 loess | 17 | 1.44 | 0.88 | 2.7 | 25.34 | 19.61 |

The particle size distribution of the loess sample in the area was also tested by a laser particle size analyzer (Type: Bettersize2000) to reflect the relationship between the static liquefaction and the particle size distribution. The particle distribution curve is illustrated in Figure 3. It can be observed that the silt (size: 0.005–0.05 mm) is the primary grain size which occupies about 67%. While the sand (size: 0.05–2 mm) content and clay (size: <0.005 mm) content are 23% and 10%, respectively. It can be regarded as low-plasticity clay(CL) according to the Casagrande classification.

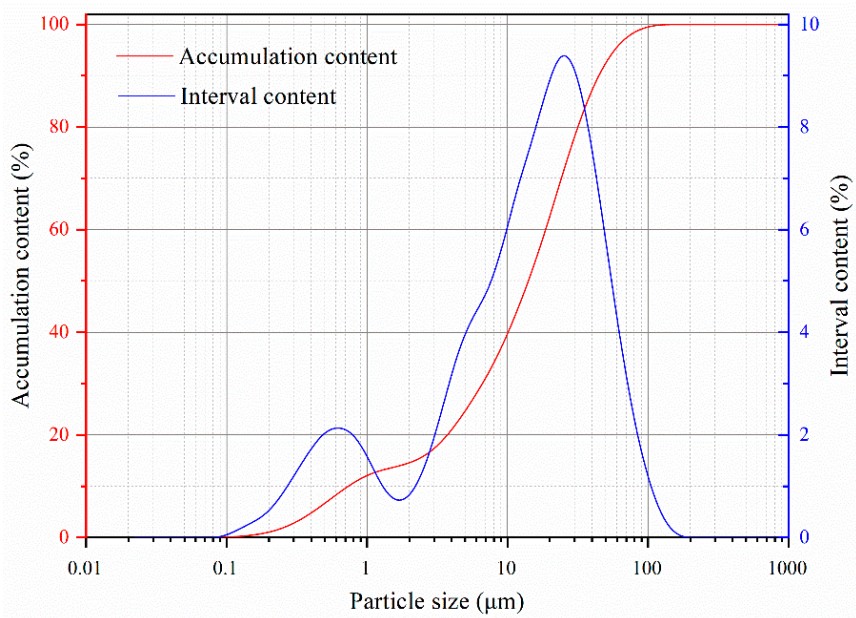

**Figure 3.** The granulometry chart of intact loess sample.

In this experiment, the undrained triaxial tests were carried out on the saturation intact loess samples by GDS triaxial instruments (Wykeham Farrance company, Hertfordshire, UK), the shear evolution characteristics of saturated loess due to irrigation water infiltration were observed.

## 3. Results and Analysis

### 3.1. Distribution Condition of Landslides on the South Jingyang Plateau

Figure 3 shows the distribution of 92 landslides on the South Jingyang Plateau from 1980 to 2016. It can be seen from the figure that the distribution of the groups of landslides in this region has the following typical characteristics:

(1) Group occurrence

The loess landslides in the area are distributed in a "bead necklace"-like pattern, which is coherently developed along the edges of the plateau but locally presents concentrated cluster features, as observed in Jiangliu, Shutangwang, Zhaitou-Miaodian, Taiping and Niujiazui. The concentration

of the landslide group is much greater than that of the surrounding areas. For example, in the Jiangliu–Miaodian area, affected by long-term irrigation and lateral erosion of the Jinghe River, there have been 64 loess landslides along the edge over a distance of nearly 10 km. Additionally, there have been 34 loess landslides along the edge of the Zhaitou–Miaodian area, which is only approximately 3.6 km long. These landslides are very dense and concentrated in such a limited platform-edge.

(2) Multiple sliding events

Loess landslides in the study area exhibit multiple failure characteristics. That is to say, due to the influence of irrigation, the same landslide undergoes many slips. Therefore, the soil around the plateau is continuously slipping, and the tableland surface continues to shrink. The multi-sequence developmental characteristics reflect the relationship between landslides in time and space to some extent, which has the following characteristics: The more times that a landslide undergoes sliding, the smaller the landslide volume that is produced, especially when the first landslide volume is large. The horizontal slip of the landslide decreases significantly with an increase in the sliding sequence. When the slope slides for the first time, the shear opening is mostly located at the slope foot and below the terraces. When the number of sliding events increases, the cutting position will gradually increase [36].

(3) Heterogeneity

The loess landslides in the study area also exhibit the characteristics of differentiation, which is mainly affected by the differences in micro-geomorphology. There are significant differences in the development types of landslides in the eastern and western sections. It can be seen from the boundary line in Figure 3 bordered by Dongzhufan Village, Taiping Town, that the eastern section is mainly dominated by landslides with long-distance flows, while the western section is dominated by collapse. By comparison, it was found that the irrigation conditions in the eastern and western sections of the plateau were the same. The regional geological background was basically the same. The slope morphology and material composition were similar, but the topographical features were obviously different. The morphology of the eastern section was relatively complete, while the western section was broken, where the gully was extremely developed. As a result, the relative drainage boundary of the surface water and groundwater in the western section was longer than in the eastern section, while the sunshine range and evaporation area were relatively greater in the western section. Therefore, long-term irrigation has less influence on the surface water and underground water level in the western tableland, which results in the differentiation of landslide development types.

### 3.2. Landslide Types on the South Jingyang Plateau

According to the statistics for loess tableland landslides in this region, the loess landslides can be divided into five types referring to the results of Hungr et al. [38]: Mudflows, clay/silt rotational slides, erosion slides, external disturbed slides, and silt topple. The corresponding distribution of these types in this region is indicated in Figure 4. Considering that this study mainly focuses on landslides, only four types of landslide development characteristics are introduced, and their representative types are shown in Figure 5 (Note: The distribution locations are marked in Figure 4). There were 92 landslides in this survey, including 35 mudflows, 37 clay/silt rotational slides, 15 erosion slides, and five externally disturbed slides (artificial loading, cutting slope toe, engineering disturbances, etc.). The proportion of landslide types is shown in Figure 6. The number of mudflows and clay/silt rotational slides in the study area is equivalent, accounting for 38.04% and 40.22% of all landslides, respectively. These numbers are far greater than the numbers of erosion slides and external disturbed slides (representing only 21.73% in total). The development characteristics of each landslide type are described below.

(a) **Mudflows:** This type of landslide mostly occurs in slope areas with abundant underground water at the foot of a slope and is the main landslide type in the area. There have been a total of 35 landslides of this type, which are significantly affected by tableland irrigation. This kind of landslide is mostly cut out from the slope foot (Q2 loess layer). Due to the high-water content of the slope foot and terrace silt layer (or sand and gravel), high pore-water pressure can be generated in the soil near the sliding surface during movement, which causes the soil in the sliding belt to fully liquefy.

Even if the terrace sliding bed is gentle, the landslides show significant characteristics of high-speed and long-distance motion. The plane shape of the sliding body is mostly semi-circular to round. The sliding distance is generally 200–300 m, with a maximum of 419 m (e.g., the Xiushidu landslide). The volume ranges from several hundred thousand square meters to one million square meters, representing medium-sized landslides. This type of landslide shows the greatest risk, the widest threat range and the most serious damage. Typical examples include the Jiangliu landslide, Dongfeng landslide, Xihetan landslide, Zhaitou arsenal landslide, and Miaodian landslide. From the perspective of the landslide period, the characteristics of flow slip mainly occur in the first sliding of the slope. However, in the local section, such as the Jiangliu landslide section, due to the accumulation of the landslide at the slope foot, the groundwater level will rise. The formation of a water seepage zone in the new slope foot section will also cause the flow slip characteristic of landslides to occur in the third and fourth phases. For example, in November 2014, January 2014, and March 2016, three consecutive third- or fourth-stage slide landslides occurred. The scale of this type of landslide is not large, but the sliding distance is far, the sliding body exhibits a high-water content, and the leading edge shows the characteristics of a mudflow.

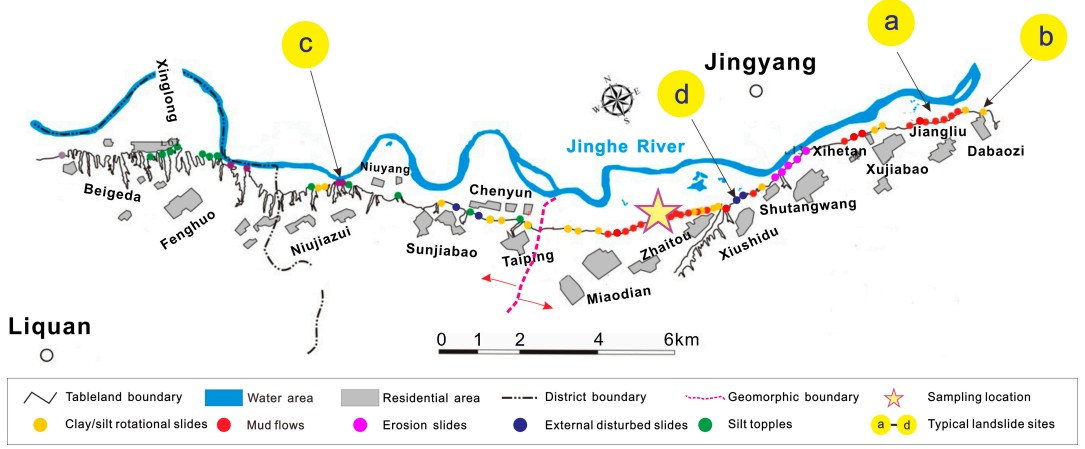

**Figure 4.** Distribution of loess landslides from 1980 to 2016.

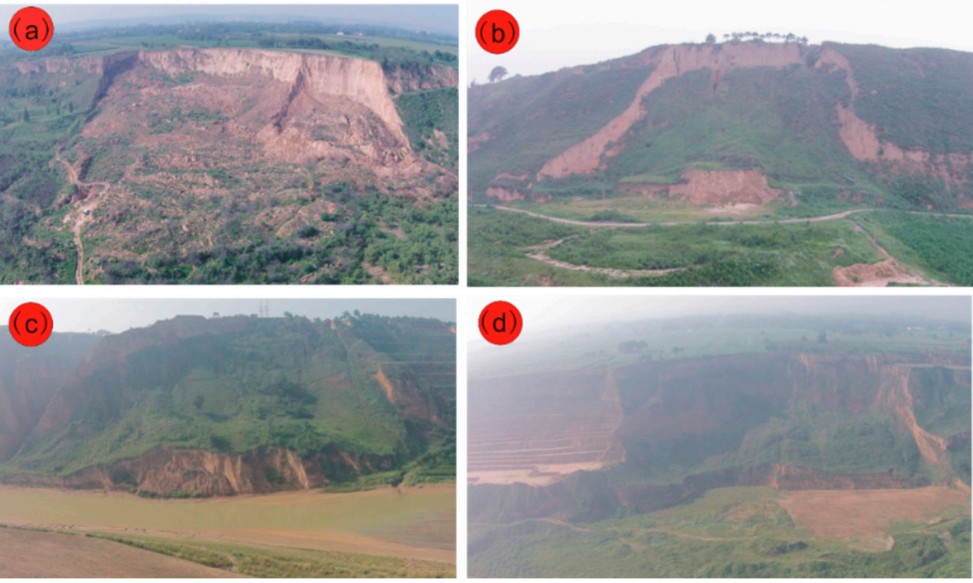

**Figure 5.** Typical types of loess landslides on the South Jingyang Plateau: (**a**) Mudflow, (**b**) clay/silt rotational slides, (**c**) erosion slide, and (**d**) external disturbed slide.

(b) **Clay/silt rotational slides:** This type of landslide is characterized by shear stress reaching the maximum shear strength of the soil and causing damage. The shear opening associated with such

landslides is relatively high and occurs in the unsaturated loess layer, which is caused by a decrease in suction in the unsaturated loess matrix due to surface water infiltration, such as that resulting from farmland irrigation. From the perspective of landslide evolution, sliding mainly occurs in the second and third landslides, and the local section corresponds to the first landslide, due to the deep groundwater. This type of landslide is generally small in scale, the slope is not saturated, and the degree of liquefaction is low during movement. The speed and distance of landslide movement are much smaller than those of the landslide, and there is little damage.

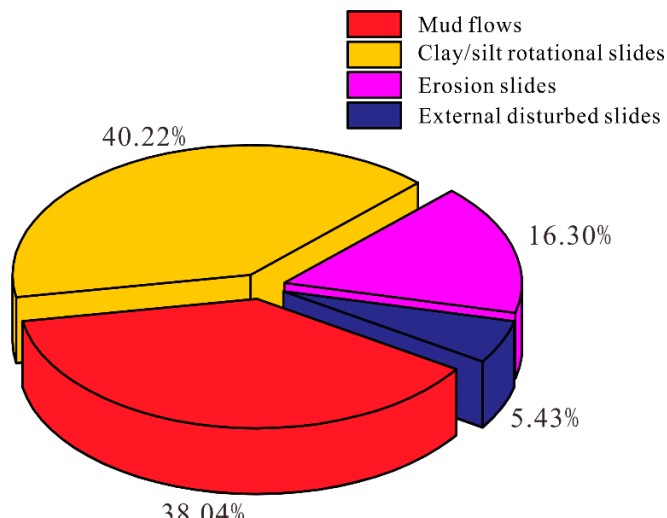

**Figure 6.** Statistics of the proportions of landslide types on the South Jingyang Plateau.

(c) **Erosion slides:** There are 15 erosion slides. This type of landslide including 15 subclasses mostly develops near the edge of the Jinghe River. Due to the lateral erosion of the Jinghe River, the slope along the edges of the plateau is steep, and the stress at the slope foot is concentrated. Due to the influence of river water infiltration, the saturation intensity of the soil at the slope foot is reduced. The slope is cut from the deep part of the slope to form a slip. Such landslides are generally thick, making them sub-stable within a certain period of time. Such loess landslides generally do not cause casualties, but they exhibit the characteristics of short incubation periods and frequent occurrences, which result in loss of land resources. These events may cause partial blockage of the Jinghe River and flooding of fields as well as affecting residents' travel.

(d) **External disturbed slides:** This type of landslide is closely related to human engineering activities. There have been five external disturbed slides in the area, affected by a cutting foot or large-scale engineering construction disturbance (such as road construction under tableland, or farming on the slope), combined with the impact of river erosion and agricultural irrigation. The distribution of this type of landslide is not regular and shows a clear relationship with the intensity and locations of engineering activities. Engineering disturbance may still play an important role in triggering landslides in this region.

*3.3. Changes in Groundwater Levels on the South Jingyang Plateau*

The estimated equilibrium phreatic water values in the irrigation area on the South Jingyang Plateau are listed in Table 2 according to a survey report from the Xianyang Water Conservancy Survey and Design Group from 1986. The annual average recharge of atmospheric rainfall precipitation that infiltrates into the soil on the South Jingyang Plateau is $188.7 \times 10^4$ m$^3$, while annual average excretion in the area is $220.8 \times 10^4$ m$^3$. The atmospheric recharge is much less than the excretion. Thus, precipitation alone is obviously not enough to raise the phreatic water level. Since the implementation of irrigation, the annual recharge has increased by $268.4 \times 10^4$ m$^3$, which means that approximately $236.3 \times 10^4$ m$^3$ of water is stored in the loess, not including the amount of excretion. According to the effective irrigation area, the groundwater level is increased by approximately 1 m.

**Table 2.** Estimated equilibrium phreatic water values in an effective irrigation area of 62.9 km$^2$ on the South Jingyang Plateau.

| Content | Type | Quantity * (10$^4$ m$^3$) | Proportion (%) | Total Quantity (10$^4$ m$^3$) | Storage (10$^4$ m$^3$) |
|---|---|---|---|---|---|
| Recharge | Rainfall infiltration | 188.7 | 41.0 | 457.1 | |
| | Field infiltration | 119.3 | 26.0 | | |
| | Channel infiltration | 149.1 | 33.0 | | 236.3 |
| Excretion | Discharge to Jinghe River | 111.8 | 50.0 | 220.8 | |
| | Pumping to irrigate | 50.0 | 23.0 | | |
| | Domestic water | 59.0 | 27.0 | | |

* According to the hydrogeological survey report for the South Jingyang Plateau from Xianyang Water Conservancy Survey and Design Group in 2016.

According to the variation of the groundwater level at a representative point on the South Jingyang Plateau (Table 3) [25,39,40], we can observe that: (1) In 1976, the groundwater depth was relatively great, and the elevation of the groundwater level was close to that of the riverbed, which was generally lower than that of the Jinghe River bed by approximately 1–8 m. (2) Comparing 1992 with 1976, the groundwater level was obviously elevated at the same point, the uplift was approximately 13–37 m, and the groundwater level was approximately 4–30 m higher than the riverbed. Taking the five representative points in Table 3 as a basic reference, the phreatic water level on the South Jingyang Plateau in 1992 was increased by an average of 23.2 m compared with 1976. The average annual increase ratio was 1.45 m. (3) In 2016 compared with 1992, with the exception of a small decrease in the water depth at Dabuzi Village, the other four points exhibited obvious increases, the uplift was approximately 4–24 m, and the groundwater level was approximately 19–37 m higher than the riverbed. In 2016, the phreatic water level on the South Jingyang Plateau was increased by an average of 33.6 m compared with 1976, with an average annual increase of 0.84 m. This finding reflects the fact that after 1976, water diversion was carried out from Baoji Gorge. With the continuous expansion of the irrigation area, the amount of irrigation water has increased, resulting in an increase in the supply of groundwater. The recharge is greater than the discharge, and the groundwater level has increased each year.

**Table 3.** Changes in groundwater levels at a representative point on the South Jingyang Plateau.

| Time | Location | Groundwater Depth (m) | Groundwater Level Elevation (m) | Difference Between the Ground Water Level and the Riverbed (m) |
|---|---|---|---|---|
| 1976 [25] | Yujiabao | 81.0 | 376.0 | −8.0 |
| | Jiangliu | 74.5 | 373.5 | −7.7 |
| | Dabuzi | 61.0 | 379.5 | −0.5 |
| | Miaodian | 66.7 | 387.9 | −8.6 |
| | Zhaitou | 52.0 | 389.0 | −2.0 |
| 1992 [25] | Yujiabu | 61.0 | 396.0 | +12.0 |
| | Jiangliu | 37.0 | 411.0 | +30.0 |
| 1992 [25] | Dabuzi | 33.0 | 407.5 | +27.5 |
| | Miaodian | 53.5 | 401.1 | +4.6 |
| | Zhaitou | 34.5 | 406.5 | +15.5 |
| 2016 | Yujiabu [40] | 42.0 | 415.0 | +19.0 |
| | Jiangliu | 28.83 | 419.17 | +37.97 |
| | Dabuzi | 37.37 | 403.13 | +23.13 |
| | Miaodian | 28.66 | 425.94 | +29.44 |
| | Zhaitou | 30.27 | 410.73 | +19.73 |

*3.4. Experimental Results of Loess Static Liquefaction of Irrigation-induced Landslides*

Figure 7 shows the stress-strain curve, pore pressure growth curve, effective confining pressure curve and stress path curve of undisturbed loess. It can be seen based on the indoor liquefaction tendency experiments as: (1) At the initial stage of loading, the deviatoric stress rises rapidly to a peak at a small strain. Then, as the strain increases, the deviatoric stress decreases sharply to a relatively stable value. Under different confining pressures, a strong strain softening type is observed. (2) When the strain is very low, pore pressure increases rapidly. After increasing to a larger value, the pore pressure increases very slowly as the strain increases and gradually becomes stable. (3) The effective confining pressure drops sharply at the beginning and then tends to be stable. The steady-state effective confining pressure increases with the increase of the initial confining pressure. At a steady state, the effective confining pressure is low, and the saturated loess is in an unstable state with low confinement, which is prone to plastic flows.

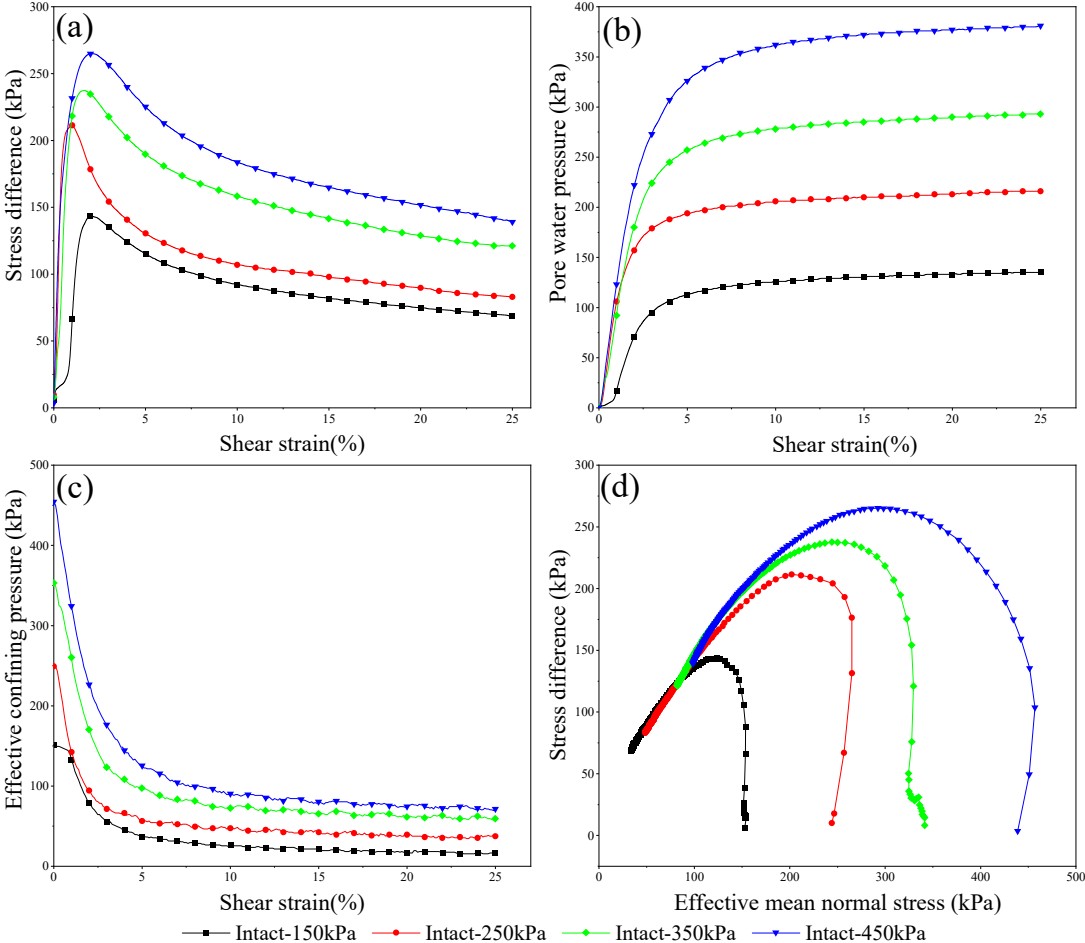

**Figure 7.** Consolidated undrained triaxial test results for undisturbed saturated loess. (**a**) Stress-strain curve of intact loess, (**b**) pore water pressure curve of intact loess, (**c**) effective confining pressure curve, and (**d**) effective stress paths.

It can be seen from Figure 7 that after the deviatoric stress reaches a peak, the soil structure is rapidly destroyed as the pore water pressure inside the soil sample continues to increase, resulting in a rapid decline in the strength of the structure. After the axial strain reaches approximately 20%, the pore water pressure and the deviatoric stress gradually reach a steady state. Based on the theory of soil plasticity, the yield surface of the specimen shrinks with strain in the stress space. Currently, the original saturated loess presents a possibility of liquefaction slippage. This possibility was verified by

experiments showing that the infiltration of irrigation water causes static liquefaction of soil saturation, which leads to landslides.

## 4. Discussion

### 4.1. Relationship Between Landslides and Irrigation on the South Jingyang Plateau

According to survey data, since diversion irrigation began in 1976 in this area, tableland margin landslides have been significantly active. From 1982 to 1985, there were four large landslides with a volume of 1.88 million m$^3$, resulting in 24 deaths, 20 injuries and the destruction of 580 mu of fruit trees. Landslide disasters were particularly prevalent from 2002 to 2006, with landslides occurring continuously in the Zhaitou–Jiangliu section, showing beading characteristics. For example, the Shutangwang landslide occurred on 2 October 2002, the Dongfeng landslide occurred on 23 July 2003, the Xiushidu landslide occurred on 12 October 2002, and the Bridgehead landslide occurred on 15 April 2006. Not only is the scale of the landslides large, their frequency is also high.

According to the statistical distribution of the landslides occurring in 2000–2017, it was found that the number of landslides and the monthly average precipitation do not show the same increasing and decreasing trends, indicating that rainfall is not the main cause of landslides in this area. There is a strong correlation between irrigation and the number of landslides. For example, the number of landslides occurring in April was highest, at 12, followed by August and March, with 10 and 9, respectively, while there were six landslides in both July and November. On the South Jingyang Plateau, March–April are the spring irrigation months. July–August are the summer irrigation months, and November is the winter irrigation month. Thus, landslides occurred most frequently in the spring irrigation period, followed by the summer and winter irrigation periods. The number of landslides outside of the main irrigation months (February, June, September, October) is smaller. These observations indicate that the occurrence of landslides in the study area is closely related to irrigation. According to local survey data, the amount of irrigation is generally highest in spring, so the frequency of landslides is also highest in this period. It can be seen that irrigation is an important trigger for landslides on the South Jingyang Plateau.

### 4.2. Analysis of Loess Landslides Development Induced by Irrigation

Based on the above-mentioned in situ survey, water infiltration due to irrigation is a significant triggering factor of inducing group-occurring of loess landslide on the loess plateau [41]. Additionally, the fissures or cracks located at the margin of the plateau provide a seepage channel for irrigation water infiltration, which in turn decreases the stability of landslides. At the same time, newly formed landslides will progressively cause new fissures, alter groundwater discharge conditions and increase groundwater levels. All these effects promote a new round of landslides. Therefore, the irrigation-associated landslides in this area present a cyclic character.

According to field survey statistics, the formation and evolution of irrigation-associated landslides can be divided into three stages.

The early stage, in which water retained in farmland after large-scale irrigation slowly migrates to the interior of the loess slope through the thick vapor zone. As the frequency of irrigation increases, the groundwater level in the area increases continuously (Table 3). The irrigation water infiltrates and erodes the loess structure in this stage, leading to continuous expansion and degeneration of vertical fissures in the loess. Continuous soil erosion results in numerous macros-cracks distributed around the top and even through the slope.

The infiltration stage of the preferential seepage channel, in which the groundwater level gradually rises with the development of the above phenomenon, the pore hydrostatic pressure increases and cracks located on the side of the plateau begin to gradually expand. Then, repeated suffusion erosion causes cracks located on the side of the plateau to develop into the preferential seepage channels for surface water infiltration, causing surface water to rapidly infiltrate the interior of the loess slopes. This

infiltration further aggravates the increase in the groundwater level, and the uplift of the groundwater level, in turn, promotes the rapid development of cracks, which causes the slope to be unstable. Additionally, according to the on-site drilling profile, there is an obvious hydraulic gradient zone close to the side of the plateau, leading to an increase in hydrodynamic pressure on the slope and a rapid increase in the development of fractures induced by hydrodynamic pressure. Furthermore, dairy irrigation is prone to remove chemical fertilizers and soil solutes, which brings about chemical corrosion and accelerates the development of sinkholes and cracks located on the side of the plateau.

The landslide sliding stage, in which surface water from irrigation continuously sinks down into the interior of the loess slopes through the preferential seepage channel, which accelerates the transfixion of the sliding surface. The water then gradually collects downwards, crossing the paleosol layer (aquifuge) at the bottom of the slope and forming perched water at the top of this layer. The bottom of the upper Lishi loess is soaked with water and gradually becomes saturated. Hence, liquefaction phenomena may occur, and the saturated softening layer is softened to be an easy-sliding layer and form as the bottom sliding surface, which brings about landslides [42].

The general process of irrigation-induced loess landslides occurring on the side of the plateau is summarized in Figure 8.

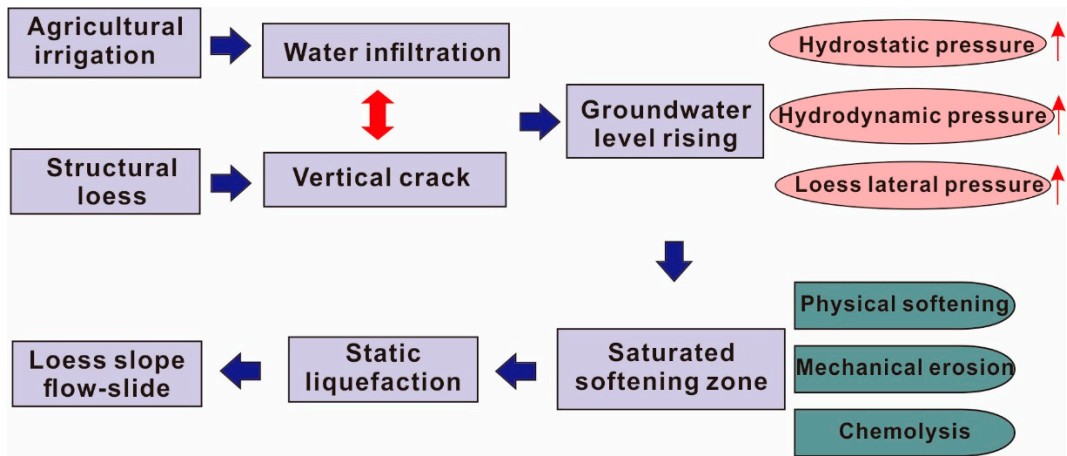

**Figure 8.** Occurrence of loess landslides due to irrigation on the side of the plateau.

*4.3. Discussion on Reasons of Landslides Induced by Irrigation on the Side of the Plateau*

Based on the site investigation, indoor tests and previous research findings, the genetic model of landslides induced by irrigation on the South Jingyang Plateau was analyzed. Three main reasons were identified:

(1) **Long-term irrigation leads to significant tension cracks on the side of the plateau.** After farmland irrigation water on the South Jingyang Plateau infiltrates the slope along the vertical joints of the macro-porous loess layer, this layer will undergo wet collapse compression under the action of the seepage force of water [43,44]. Consequently, groundwater on the plateau is slowly uplifted, crosses the paleosol layer (aquifuge) at the bottom of the slope, forms perched water at the top of this layer, and the bottom of the upper Lishi loess is soaked in water and gradually saturated. The water level continues to rise, which causes the saturated layer to thicken and the shear strength to decrease, until water basically submerges part of the sliding surface. Then, deformation failure begins to take place in the soil at the foot of the slope under the action of overburdening by pressure, which in turn causes deformation of the whole slope. Furthermore, soil stress redistribution in the slope increases the shear stress of the upper soil, and splitting failure occurs in the slope soil because shear strength is lower than shear stress. Thus, tension crack occurs at the trailing edge of the slope.

In addition, aquifuge action of the paleosol layer causes the upper loess layer to collapse and uneven subsidence to occur. This phenomenon also causes settlement to be rather great in the loess

layer inclined to the free face of the slope, while that of the loess layer on the plateau is small. Thus, the barycenter of the slope moves outwards, which will aggravate the cracks on the side of the plateau.

(2) **Continuous irrigation causes the formation of a saturated softening layer.** Continuous irrigation causes water to infiltrate vertically to the top of the paleosol layer at the foot of the slope (relative to the aquiclude) and to form a saturated layer at this location. Irrigation causes the groundwater level to rise gradually, which increases the thickness of the saturated layer accordingly, and the loess layer relative to the water-repellent layer gradually softens under the action of water, which reduces frictional resistance and forms a saturated softening zone. Moreover, soil in the saturated softening zone is subjected to the combined action of seepage water pressure and pressure overburdening, further destroying its original structure, while soluble salt partially dissolves under the action of water, which aggravates sludging of the sliding zone soil and reduces the frictional resistance of the sliding zone soil.

(3) **The saturated softening zone is prone to flow slides after static liquefaction occurs.** With increasing irrigation, groundwater in the plateau area gradually rises and reaches a certain height, which results in a large hydraulic gradient. This gradient, in turn, leads to an increase in the thickness of the saturated softening zone at the foot of the slope. As the shear strength of the undisturbed loess is significantly reduced and liquefaction potentially occurs, the softening zone of the sliding zone soil gradually penetrates the slope, which causes a flow slide [32,44]. Then, the relative sliding between the loess particles in this layer causes pore pressure to increase sharply, directly leading to reduction of the effective stress in the slope softening zone and finally bringing about a flow-slide type landslide induced by liquefaction.

## 5. Conclusions

The types and developmental characteristics of loess landslides around the South Jingyang Plateau were first determined. Then, the seasonal agricultural irrigation features and groundwater changes were introduced. Clear evidence was found that irrigation is a significant factor influencing changes in groundwater and even causing the occurrence of landslides. Based on a field investigation and indoor experiment, the intrinsic mechanism of shallow loess landslides induced by seasonal agricultural irrigation was studied in detail. Several main conclusions can be reached.

The shallow loess landslides around the South Jingyang Plateau exhibit three typical characteristics: Group occurrence, multiple times of occurrence, and temporal heterogeneity. Furthermore, they can be divided into five types: Mudflows, clay/silt rotational slides, erosion slides, external disturbed slides and loess topples, among which, mudflows and clay/silt rotational slides account for approximately 78% of the total number of landslides investigated. Seasonal agricultural irrigation on the South Jingyang Plateau can be divided into three stages: Spring irrigation, summer irrigation, and winter irrigation. Current agricultural irrigation in this region is mostly based on traditional flood irrigation, which brings about obvious uplift of the groundwater level in the study area. According to survey data, the occurrence of plateau margin landslides in the study area is closely related to agricultural irrigation.

The formation and evolution of irrigation-associated landslides on the South Jingyang Plateau can be divided into three stages: The early stage, the infiltration stage of preferential seepage channels, and the landslide sliding stage. During this process, the occurrence of landslides is closely related to loess softening, mechanical subsurface corrosion and chemical corrosion caused by groundwater level uplift, and static liquefaction instability finally occurs in the softening zone of loess. Similarly, an indoor experiment confirmed that undisturbed loess under different confining pressures shows a strong stress-softening characteristic. This characteristic is more significant under the condition of low confining pressure, like the "static liquefaction". The genetic model of landslides induced by irrigation on the South Jingyang Plateau consists of three main factors. Long-term irrigation leads to significant tension cracks on the side of the plateau. Continuous irrigation causes the formation of a saturated softening zone, and the saturated softening zone is prone to flow slides after static liquefaction occurs.

Based on the above research, the key to controlling the landslides around the South Jingyang Plateau lies in limiting the recharge–discharge relationship between irrigation water and groundwater. Suggestions for improvement measures are as follows: Move away from the traditional agricultural irrigation method of flood irrigation, and advocate for drip irrigation or well irrigation by pumping groundwater. It is conceivable to carry out grouting and recirculation treatment of cracks on the side of the plateau, but the selection of slurry must take into account ecological and environmental issues. Adopt the rainfall pattern of drilling horizontal holes to reduce the groundwater level at a portion of the plateau at the foot of the slope. Pay attention to the strict inspection and timely repair of irrigation canals.

**Author Contributions:** R.-X.Y. and J.-B.P. conceived and designed the survey and experiment; R.-X.Y., L.-J.C. and C.-Y.K. performed the field survey and experiments; J.-B. Peng, Q.-B.H. and Y.-J.S. contributed funding supports; R.-X.Y. and Y.-J.S. wrote the paper. J.-B.P. revised the English of this paper.

**Funding:** This work was supported by the Major Program of National Natural Science Foundation of China (Grant No. 41790441), the National Natural Science Foundation of Shaanxi Province, China (Grant No. 2018JQ5124) and the Foundation of Key Laboratory of Western Mineral Resources and Geological Engineering of Ministry of Education, Chang' an University (Grant No. 300102268503).

**Acknowledgments:** We thank the entire team for their efforts to improve the quality of the article. At the same time, we would like to thank editor for his timely handling of the manuscripts.

**Conflicts of Interest:** The authors declare no conflict of interest.

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
