# Peer review of "Triggering Influence of Seasonal Agricultural Irrigation on Shallow Loess Landslides on the South Jingyang Plateau, China"

_water, doi:10.3390/w11071474_

Round 1

Reviewer 1 Report

The manuscript (MS) deals with a mainly descriptive analysis how irrigation practices may have affected the occurrence of shallow loess landslides in the South Jingyang Plateau in China, starting from 1980s. English is acceptable (just some minor edits are needed). Given the descriptive character of the MS, perhaps it more suitable for MDPI Geosciences journal instead of MDPI Water.  The MS presents two main methodological issues as I illustrate in tthe "Specific comments" section. Given the importance of these issues, I suggest major revisions for the MS.

Specific comments

Table 1: This hydrological balance seems uncorrect. Groundwater recharge cannot be due to both rainfall and infiltrated water. In fact only the proportion of rainfall that infiltrates into the soil should be considered, the other going to runoff. It is unclear if this proportion is included in "field infiltration" which proportion of "rainfall" goes to runoff. This affects also text of section 3.2 and data presented in Table 2.

Figure 5 and related text. The authors state throughout the MS that landslide occurrence is strictly related to irrigation practices. Indeed they do never prove this, but just discuss this as it sure this is the case. They should compare landslide occurrence data before and after 1980s when irrigation had a shift in the area. If different frequencies of landslide occurrence are seen in the two periods then it is more evident that irrigation is a main cause.

Author Response

Dear Reviewers:

Thank you for your professional opinions and pertinent evaluations concerning our manuscript entitled “Triggering Influence of Seasonal Agricultural Irrigation on Shallow Loess Landslides on the South Jingyang Plateau, China”. We have carefully considered your constructive and valuable comments and revised the manuscript accordingly. Corrections and changes are printed in BLUE in the revised manuscript.

Reviewer 2 Report

Dear authors, 

This is an interesting and well written paper where the influence of irrigation on landslide occurrence in Jingyang is investigated. 

Please make minor changes that are written in comments. 

Author Response

Dear Reviewers:

Thank you for your professional opinions and positive comments concerning our manuscript entitled concerning our manuscript entitled “Triggering Influence of Seasonal Agricultural Irrigation on Shallow Loess Landslides on the South Jingyang Plateau, China”. We have carefully considered your constructive and valuable comments and revised the manuscript accordingly. Corrections and changes are printed in BLUE in the revised manuscript.

Reviewer 3 Report

The research focus on landslides occurrence in a loess area resulting form interaction of human actions, that is irrigation for agricultural, with natural processes. The topic is interesting but the paper in its actual version is affected by a major problem and some minor ones.

At first it must be restructured as actually general outline of the area, data, findings from previous researches and new findings are mixed together. The paper should be written using the typical structure of a scientific paper, that is with the following paragraphs: 

Introduction, that should introduce the problem, starting from a general view; agriculture and stability is an issue all over the world, in different contexts as for example in terraced landscape (see appropriate papers in literature);

Materials and methods - comprising the outline of the study area (geographic, geologic-geomorphologic, climatic) and the description of the methodology adopted,

Results, that must describe the findings according to the methodology;

Discussion, that should discuss the results;

Conclusions, that should not be a mere repetition of what has been previously written as in the present version of the paper; it should address solution and discuss them more deeply than only in a couple of sentences.

In the actual version it is not clear what are the new data and findings effectively obtained, results are spread in many parts of the papers, while they need to be clearly differentiated from previous findings etc.

Besides, the outline is actually not sufficient: the map is poor and lacks in a location scheme that should allow the reader to understand where the study area is. The actual scheme does not allow to clearly identify the elements and the morphological context. The samples used for laboratory tests should be located on the map and should be specified at least how many they area and at what landslide type they belong to. The actual description is not sufficient.

The landslide types should be described and classified according to the literature (Varnes, Hungr) and in general references appears too limited in particular a more international and wider view is needed.

Some minor issues are: the labels in all the figure need to be magnify for improving the readability; english language needs to be improved.

Finally there is space enough to add at least one figure to improve the paper.

Some annotations are in the attached file.

Author Response

(The authors gave the same response as above.)

Round 2

Reviewer 1 Report

I see that the paper has been improved. I suggest to revise further the language. For instance the word "issued" should be replaced with "stated" in many points.

The paper could benefit from a sentence mentioning the possible interaction of climate change with landslides, integrated with some literature in the subject:

M. Alvioli, M. Melillo, F. Guzzetti, M. Rossi, E. Palazzi, J. von Hardenberg, M. T. Brunetti, S. Peruccacci IMPLICATIONS OF CLIMATE CHANGE ON LANDSLIDE HAZARD IN CENTRAL ITALY,  Science of the Total Environment 630, 1528-1543 (2018)

GARIANO, Stefano Luigi; GUZZETTI, Fausto. Landslides in a changing climate. Earth-Science Reviews, 2016, 162: 227-252.

Peres, D. J., & Cancelliere, A. (2018). Modeling impacts of climate change on return period of landslide triggering. Journal of Hydrology, 567, 420-434.

Thus the paper can be accepted after minor revisions.

Author Response

Thank you for your professional opinions and pertinent evaluations concerning our manuscript entitled “Triggering Influence of Seasonal Agricultural Irrigation on Shallow Loess Landslides on the South Jingyang Plateau, China”. We have carefully considered your constructive and valuable comments and revised the manuscript accordingly. Corrections and changes are printed in BLUE in the revised manuscript.

Reviewer 3 Report

The paper has been improved and a general restructuring of sections have been done, but the paper still needs some corrections.

The authors, in their replay to the first review, refers to a Materials and methods section but I do not find it (replay to lines 285-324). Probably it could be paragraph number 2, but its title is different and partially redundant with sub-paragraph. Authors can rename it as Materials and method and leaving to the subparagraphs titles the relative description. The complete methodology should be explained here, comprising the description and source of all the data used in the research.

The introduction, as already sad in the first review, would gain in appeal if the work is put into the more general context of the relationships between man activities (man induced processes) and natural processe. I made the example of man made terraces because they are related to agricultural practice: terraces abandonment in several areas of the world represents actually a threat and they are regarded as a possible source of instability. Other examples could be done.

As already asked, please specify how many soil samples have been analyzed.

Many issues regard the sequence number of figures and tables: please revise accurately.

The location scheme that has been added to the old figure n.1 must be the first figure as, at first, one should know where the place you are speaking about is.

Please find in attachment some other notations.

Author Response

Thank you for your professional opinions and positive comments concerning our manuscript entitled concerning our manuscript entitled “Triggering Influence of Seasonal Agricultural Irrigation on Shallow Loess Landslides on the South Jingyang Plateau, China”. We have carefully considered your constructive and valuable comments and revised the manuscript accordingly. Corrections and changes are printed in BLUE in the revised manuscript.
